# Understanding Motivations for Plural Identity on Facebook among Nigerian Users: A Uses and Gratification Perspective for Engaging on Social Network Sites (SNS)

**Tawfiq Ola Abdullah** [1,*], **Brent J. Hale** [1] **and Mutiu Iyanda Lasisi** [2]

1   School of Media & Communication, University of Southern Mississippi, Hattiesburg, MS 39406, USA; brent.hale@usm.edu
2   Department of Media, Faculty of Creative Industries, National Research University Higher School of Economics, 101000 Moscow, Russia; mutiu.iyanda@gmail.com
*   Correspondence: tawfiq.abdullah@gmail.com

**Abstract:** In the context of the increasing proliferation of users on social networking sites (SNS) and the ensuing debate on their benefits and drawbacks, this study examines the interconnection between human behaviors and identity formation on Facebook. We leveraged the concept of plural identity, seeking to identify relationships between online social behaviors and plural identity tendencies. We conceptualize plural identity as a construct spanning the personal and social dimensions of identity, and use these as the core starting points for studying plural identity. Accordingly, the relationships between social-communicative and personal-communicative behaviors involving plural identity on Facebook were investigated. A survey administered to Nigerian Facebook users (N = 429) revealed that social-communicative behaviors (i.e., social support and social interaction) exhibited strong relationships with plural identity on Facebook; similarly, personal-communicative variables (i.e., presentation of the extended self and self-expression) were strongly related to plural identity. This study highlights the role of SNS in satisfying peoples' social and communication needs, which are interwoven with identity formation.

**Keywords:** plural identity; social media; online behavior; Facebook; Nigeria

## 1. Introduction

As conceptualized by Sen (2007), plural identity is a way to understand a person's identity. He defines the concept by listing various categories to which a person may concurrently belong. For example, a person may be a husband, a father, a scholar, an educator, a Christian, an activist, and an American citizen. He berates the widespread practice of cramming people into boxes of singular identities, such as religion-based characterizations, especially among politicians and policymakers. Such singular characterizations lead to erroneous local, national, and international policies which are noxious to society. Sen (2007) contends that a unidimensional view of a person's identity arises either from inexperience with diversity or from deliberate disregard for it.

According to Schaetti (2015), people express identity in popular culture daily, making identity a familiar concept to nearly everyone. However, it is multilayered and complex. It has attracted interest from researchers and educators, especially in psychology and sociology, and has served as a basis for decision-making by politicians, market forecasters, and advertisers (Sen 2007). To an individual, creating awareness of one's identity is fundamental to developing intercultural competence. Schaetti (2015) defines identity as "a person's largely unconscious sense of self, both as an individual and as part of the larger society" (p. 410). It influences with whom people affiliate and from whom they de-affiliate, why and how they view the world the way they do, and to what degree they are willing to engage people different from themselves.

In addition to the intercultural competence and socialization that understanding identity (both of self and of others) enables, it enables communication as well. According to the Communication Theory of Identity (CTI), individuals internalize social relations and role identities, which are enacted through communication, that is, in relation with others (Hecht et al. 2005). Put differently, identity reflects social roles and relations, and these behaviors concurrently redefine an individual's self-identity. Furthermore, social behavior, like social interaction, is realized through communication according to relevant identities and roles associated with different contexts. CTI concentrates on the two-way interaction between identity and communication, conceiving identity as communication, instead of construing identity as a mere product of communication or vice versa, e.g., (Hecht 1993). Stemming from this communicative perspective, the present work assesses individuals' plural identity on social networking sites (hereafter referred to as SNSs). SNSs are a subset of social media characterized mainly by enabling user-generated content, sharing of messages, and socialization with predominantly offline acquaintances (Kaplan and Haenlein 2010). Particularly prominent examples include Facebook (the focus of this study), Twitter, and Instagram.

There has been previous research into how people create their identities and manage them offline and online, however, there has been little, or no attention paid to how they negotiate their plural identities on SNSs. Understanding how people navigate their identities on SNSs, especially considering the complex relationship between personal and social aspects is an important research gap, particularly given the ever-increasing movement to and presence of people on SNSs. As of July 2022, the active global social media population was 4.7 billion (Statista 2022), approximately 59% of the estimated global population of 8 billion (Pew Research Center 2022). Users' inclination towards SNSs has increased exponentially, with users engaging with these platforms by sharing life activities or expressing their emotions about events such as sports, crises, and more (Agarwal and Toshniwal 2019). For example, DiMicco and Millen (2007) found that the use of Facebook is permeating the workplace and becoming part of the daily routine of young hires. They envisaged that SNSs would likely become an integral part of many workplaces as they became more popular. With the rising importance of SNSs to people's daily routines and the permeation of SNSs into many facets of daily life, identity manifestations on these platforms should be studied in more details. This study contributes to the scholarship examining how people negotiate their identities on SNSs, an integral part of contemporary human life. Accordingly, this study seeks to understand the relationship between communicative behaviors and plural identity on Facebook, with a focus on Nigerian users.

## 2. Literature Review

### 2.1. Identity and Plurality

Identity has been described as "ambiguous and slippery", as researchers have overused it in different contexts and for profuse and manifold purposes, according to Buckingham (2008, p. 1). He explains further that the term is fundamentally and inherently paradoxical, simultaneously suggesting similarity and difference. On the one hand, identity is a consistent and enduring term unique to each individual. On the other hand, it references an individual's relationship with another individual or group, suggesting interconnectedness between persons and groups. Because identity is notoriously amorphous, this study adopts and focuses on the personal and social types of identity, even though scholars have proposed other types of identity in addition to these two (e.g., Goffman 2009; Hecht et al. 2003). Thus, the plurality of identity is conceived through the combination of the personal and social identity components.

Personal identity consists of general beliefs about one's aspirations, goals, and values (Erikson 1968; Marcia 1966), including personal conceptualizations about one's physical, psychological, and social features and abilities (Harter 2015). Personal identity includes the dynamics of mental health as well as the variety of interpersonal abilities people bring to their lives (Schaetti 2015). Conversely, social identity refers to "the individual's

knowledge that he/she belongs to certain social groups together with some emotional and value significance to him/her of the group membership" (Tajfel 1978, p. 31). People often confound these two dimensions of identity, although they share no overall pattern despite their close relationship. For example, while people's sense of belonging (social identity) can bring about positive feelings about themselves (personal identity) when their group experiences good things, self-esteem (personal identity) is not directly caused by reference group orientation (social identity). Underlying this, it is important for people to have a sense of belonging; thus, which reference group they orient to matters less than satisfying this basic need (Schaetti 2015). Additionally, people can orient to more than one group, creating a complex web of interacting identities.

Researchers have largely agreed that self-identity is pluralistic, not singular. Out of the many recent developments in understanding the social nature of the self, probably the most notable has been the theoretical distinction between personal and social identity, initially advanced by Henry Tajfel and John Turner as constructs of social identity theory (SIT) and later broadened in self-categorization theory (Tajfel and Turner 1979; Turner et al. 1987). This latter perspective of the self is conceptualized as a hierarchical structure consisting of strata of graduating abstractions that each add to an individual's sense of who they are. At the interpersonal stratum of the system, researchers describe personal identity by differentiating the individual from members of the in-group (e.g., "I'm a unique character, imaginative, distinct, and original"). At the intergroup stratum, an individual's characterization comprises social identities with emphasis on stereotypical similarities shared among group members (e.g., "I'm an African, a philosopher, an Atheist"). The ascription of these social identities may be from birth—like gender and race—or may stem from one's membership in groups. However, group membership alone is not a sufficient category for self-identification; recognizing and accepting one's membership as self-defining is the decisive criterion for self-identification (Brewer 1991; Deaux 1992; Turner 1984). In essence, these researchers recognize that a person's identity constitutes both personal and social aspects, and that self-identity is socially constructed.

Furthermore, psychologists have identified that the "self" represents more than just a collection of personalized attributes that remain constant over time and across contexts (McGuire and McGuire 1982; Gergen 1981; Fazio et al. 1981; Niedenthal and Beike 1997). Contemporary perspectives on the self's structure and content include the broader constructs of culture and collective, defining the self as relating oneself to others (Clarke 2008; Oyserman 1993; Deaux 1992; Brewer 1991; Moscovici 1984; Tajfel and Turner 1979; Triandis 1989, 1990). In addition to the claim that a person's self-identity is pluralistic and socially constructed, it is dynamic and varies according to the context in which individuals find themselves.

The determinants behind which of these identities is operative at a given time are a combined function of stable and dynamic forces (Kondo 1990; Markus and Wurf 1987; Markus and Kunda 1986). Regarding the self-categorization tradition, researchers emphasize the intrinsic and varying nature of identity and its dynamic movement within the changes of the comparative context (Turner et al. 1994). According to Nario-Redmond et al. (2004), situational variability is significant in personal and social identity salience as well as dispositional tendency toward self-definition at the levels of personal or social categorization. They argue "that the importance assigned to either of these two levels helps to define the self across situations, as perceivers actively select self-categorizations that are central, relevant, and useful" (p. 144). They further maintain that individual differences exist between the centrality and importance of the personal and social identity spheres. Essentially, the self-identity that individuals decide to present, whether at the personal or social stratum of identity, is determined by the salience of the situation or context involved and that individual's disposition at a particular time.

## 2.2. Theoretical Basis and Existing Empirical Studies on Plural Identity

Other theories establish the plurality of self-identity, including Identity Negotiation Theory (INT) (Ting-Toomey 2015), which proposes that an individual's identity consists of multifaceted components of culture, ethnicity, religion, social class, gender, sexual orientation, profession, family/relational role, and personal image(s) built on self-reflection and other social categorizations. According to social identity theory (SIT) (Tajfel 1978), social or socio-cultural identities can include ethnic affiliation and familial roles. Personal identities can include any distinctive attributes related to oneself compared to those of others. Hence, everyone's composite identity considers group membership, relational roles, and individual self-reflexive components. Individuals primarily obtain their composite identity via social conditioning processes, individual lived experiences, and repetitive intergroup and interpersonal interaction. Irrespective of whether an individual is or is not conscious of these identities, identity self-conception and other typecasting impact individuals' daily behaviors in both generalized and particularized manners.

In explaining CTI's embracing of personal and social relations as loci of identity and the close association between identity and communication, it posits four loci or frames of identity: (1) personal, (2) relational, (3) enacted, and (4) communal (e.g., Hecht et al. 2003). These loci of identity interpenetrate one another, and individuals often experience identity gaps in communication across these loci of identities (Jung and Hecht 2004). Additionally, the four identity frames are sometimes inconsistent with one another (e.g., through contradiction or exclusion). Despite the conflict, they cohabit and collaborate as components of identity. Put differently, the four frames interact in a dialectical tension, and the identity gap is a dimension of this tension. "Identity gaps are defined as discrepancies between or among the four frames of identity" (Jung and Hecht 2004, p. 268), and can exist between and among any of the identity foci.

The implication of the above is that multifaceted and multilayered identity is universal and underscores the plurality of identity. Scholars across various traditions have theorized about the plurality of humans' identities. Accordingly, self-identity can be identified and defined as a combination of personal identity and social identity domains. When these different domains are isolated, each has different layers of identities for an individual. Moreover, when a self-identity is compared across the personal and social domains, identity is pluralistic both theoretically and practically. This study expects that the tendency for individuals to define and perform their identities across personal and social spaces is not only realizable offline (i.e., the unwired world), but is achievable online (i.e., the virtual world) as well, taking into consideration the pervasiveness of social media in modern societies.

## 2.3. Use and Gratification Theory and Related Works

Use and gratification theory (hereafter referred to as U&G) is based on the social and psychological origins of needs, which generate expectations of the mass media or other sources, in turn leading to varying patterns of media exposure and resulting in need gratification and other consequences (Katz et al. 1973). U&G theory postulates that the social and psychological effects of internet use depend on the user's reasons and goals for using the technology (Weiser 2001). Shao (2009) claims that individuals embrace user-generated media for different purposes, for example, consuming content to meet their information, entertainment, and mood management needs, participating by interacting with content to promote social connections and virtual communities, and producing unique content for the purposes of self-expression and self-actualization. Colás Bravo et al. (2013) confirms that "for young people, online social networks are a source of resources used to fulfill needs, both psychological and social" (p. 20).

People display different motivations for using social networks, which can be classified as seeking information, seeking entertainment, engaging in social interaction, developing personal identity, and self-disclosure (Omar et al. 2014). According to McQuail (1987), seeking information is a basic need for people to understand relevant events and conditions,

while seeking entertainment is a means of destressing and relaxing as well as of occupying leisure time. Social interaction and integration involve gaining insight into others' conditions or social empathy. Finally, Jourard (1964) identifies self-disclosure as a way of making a person transparent to others and helping others see a person as a distinctive or admired human being (which is another potential motivation for using SNSs).

Other studies have investigated motivations in the context of the internet and SNSs (Stafford et al. 2004), including socialization and entertainment (Curras-Perez et al. 2014), loneliness (Petrocchi et al. 2015), social integration (Weiser 2001), self-expression and self-actualization (Shao 2009), and work–life balance (Nam 2014). Scheepers et al. (2014) included more motivations for SNS use: information seeking, hedonic activities, maintaining strong ties, and expanding weak ties. Another study about Turkish students found that behavioral intention to use SNSs on smartphones included perceived enjoyment, perceived ease of use, perceived usefulness, and social influence, with each exhibiting either direct or remote effects on SNS use via smartphone (Calisir et al. 2013). In sum, people have varying incentives for employing technology, including SNSs. Regarding media and identity, personal identity entails reinforcing personal values, finding behavioral models, identifying with other values (in the media), and gaining insight into oneself (Omar et al. 2014). Hence, there is a need to explain the communicative behaviors that may relate to plural identity formations on SNS.

### 2.4. Communicative Behaviors on Social Media

Communicative behaviors on social media are diverse and have attracted scholars' attention with a focus on both formal and informal contexts, with a common interest in behaviors that are peculiar to specific digital platforms. In organizational settings, for example, individual, interpersonal, and organizational-level behaviors have been found to underly employees' communicative behavioral (ECB) intentions on social media (Lee 2020). In informal settings, such as in a crisis, for example, people's concern about a public emergency and the need for information surveillance by governments are among the behaviors that have been found to influence social media use (Xie et al. 2017). With respect to the present study more specifically, plural identity manifestation on Facebook may relate to various communicative behaviors, which may be categorized into personal-communicative behaviors (i.e., presentation of extended self and self-expression) and social-communicative behaviors (i.e., social obligation, social support, and social interaction).

### 2.4.1. Presentation of the Extended Self

Belk (1988) asserted that humans consciously or subconsciously, and deliberately or inadvertently, regard their possessions as part of their selves. Individuals' self-definition consists of their inner cores and aggregate selves, including family, neighborhood, and nation. There are diverse possessions that have different degrees of importance to a person's sense of self and enabling self-constructions. The major classifications of extended self are the body, internal processes, ideas and experiences, things, places, and people an individual feels attached to. However, the extended self has been reexamined in response to emerging digitalization and with consideration of social media and other technological devices, resulting in the transformation of the concept of the extended self from primarily corporeal entities to include virtual entities (Belk 2013). Kang and Shin (2021) investigated the mediated relationship between the use of Facebook as an extended self and users' privacy management online. They found that motivations of self-expression, belonging, and archiving personal memory were indirectly associated with privacy management. Their findings underscore the proposition that self-extension with respect to Facebook is a vital psychological construct determining how various self-related incentives impact users' privacy management.

### 2.4.2. Self-Expression

Self-expression is the expression of one's thoughts and feelings through words, choices, and actions (Kim and Ko 2007). It is supported by the constitutional provision of freedom of expression in democracies, and its social and psychological consequences have been scientifically demonstrated (Kim and Ko 2007). For example, previous works have found a positive connection between self-expression and people's psychological state (Pennebaker et al. 1988; Freud 1989). However, despite the universality of self-expression, it is culturally divergent (Kim and Ko 2007; Kim and Sherman 2007; Kim and Drolet 2003), with Western cultures regarded as largely individualistic and encouraging of self-expression, while in contrast, Eastern Asian culture is predominantly collectivistic and discourages self-expression.

**RQ1:** *What is the association between the use of Facebook for (a) presentation of the extended self and (b) self-expression and plural identity manifestation?*

### 2.4.3. Social Obligation

Scholars have studied social obligation in organizational contexts, referring to it variously as corporate social responsibility and as encompassing organizations' relationship with employees and their external environments (e.g., Windsor 2001; Moir 2001). It refers to any law, regulation, directive, or code of practice that a company or its subsidiaries uses in governing the relationship between them and their respective employees (Law Insider n.d.). However, social obligation belongs to the larger society, and emanates from the family unit. In this sense, social obligation practices have cultural differences similar to self-expression (Janoff-Bulman and Leggatt 2002).

### 2.4.4. Social Support

Social support refers to both sociopsychological and tangible resources at individuals' disposal through their interpersonal relationships (Rodriguez and Cohen 1998). It exerts beneficial effects on the mental and physical health of the recipient. Social support generally stems from the processes individuals undertake in managing the resources provided by their social networks to enhance coping and recovery from distress. With the advent of the internet and social media, people are expanding their social networks to virtual worlds, and scholars have shown interest in studying how people practice social support on SNSs (e.g., Oh and Syn 2015; De Choudhury and Kiciman 2017).

### 2.4.5. Social Interaction

Social interaction refers to encounters between a minimum of two people in which they give attention to one another and regulate their behaviors in consideration of each other (Reis et al. 1980). Hoppler et al. (2022) identify six components of social interaction: actor, partner, relation, activities, context, and evaluation (APRACE). Scholars have identified interpersonal social interaction as one of the motivations for using social media (e.g., Whiting and Williams 2013; Fischer and Reuber 2011). However, Cerulo (2011) has drawn the attention of scholars beyond human-to-human social interaction, elucidating the development of other forms of social interactions (e.g., human–robot communication), resulting in another research area (e.g., Nass and Moon 2000; Nass et al. 1996).

**RQ2:** *What is the relationship between the use of Facebook for a) social obligation, b) social support, and c) social interaction and plural identity manifestation?*

### 2.5. Social Media and Identity

Investigating people's motivations for social media use, at least those relevant to plural identity, involves the self, communication of personal identities, and social exchange. Identity is a complex facet of human nature that can be understood through five interlocking factors: (1) level of dissociation and integration, (2) positive and negative valence, (3) level

of fantasy and reality, (4) level of conscious awareness and control, and (5) the chosen media (Suler 2002). In addition, DiMicco and Millen (2007) suggest that people manage their online identities by combining their offline and online friends on Facebook during transitions (e.g., from college to the workplace). This study investigates both personal and social-oriented motivations resulting in plural identity on Facebook. First, the internet offers people unprecedented opportunities to present themselves in various ways through the information they make available online (Suler 2002). Put differently, people are able to engage in personal-communicative behaviors (e.g., presentation of extended self and self-expression) to construct their individual identities in online spaces.

In addition to the prediction that people use social media for personal identity gratification, this study expects that people will exhibit stronger motivations to use SNSs for social-behavioral incentives (Kaplan and Haenlein 2010). This higher premium on social-behavioral incentives could emerge because personal and social identities are socially constructed and evolve through interaction with others. Additionally, social media platforms generally encourage interaction with an amalgam of people, with the primary purpose of socialization. Therefore, socially oriented behaviors that users may manifest on social media could have important implications for the emergence of plural identity via the strengthening of social identities. In summary, people have personal considerations and different identity-related motivations for using SNSs, which are in turn configured to meet the different motivations that people bring to their use of these sites.

## 3. Method

Considering that people express plural identities offline, traversing the intrapersonal, interpersonal, and intercultural communication contexts, this study investigates the relationship between personal and social behaviors and the plurality of identity displayed by Nigerian Facebook users. A survey was conducted to accomplish this aim, and details are provided in the following sections.

### 3.1. Participants and Data

Nigerian Facebook users were recruited to participate in a survey (N = 429) between July 2021 and January 2022. Overall, after data cleaning 332 valid responses were collected using an online survey, while an additional 97 responses of Nigerian Facebook users were collected offline using hard copies of the questionnaire. The respondents' demographics are listed in Table 1. Nearly every respondent was an active Facebook user (53.6% of respondents participated in at least five Facebook groups); respondents were largely 18–45 years old (88.8%) and college educated (66.4% had earned bachelor's degrees), and men were slightly overrepresented in the sample (56.2% of respondents). The respondents included both diasporic Nigerians and Nigeria-domiciled citizens.

**Table 1.** Respondent demographics (N = 429).

| Item | Characteristics | Frequency | % |
|------|-----------------|-----------|---|
| Nationality | Nigerians | 429 | 100 |
| Active Facebook user | Yes | 429 | 100 |
| Belonging to Facebook groups | Yes | 408 | 95.1 |
| | No | 21 | 4.9 |
| Number of Facebook groups | 1 | 39 | 9.1 |
| | 2 | 28 | 6.6 |
| | 3 | 48 | 11.2 |
| | 4 | 20 | 4.7 |
| | 5 or more | 292 | 68.3 |
| 18 years old or above | | 429 | 100 |
| Age | 18–30 | 177 | 41.3 |
| | 31–45 | 196 | 47.5 |
| | 46–60 | 49 | 11.4 |
| | 61+ | 7 | 1.6 |

**Table 1.** *Cont.*

| Item | Characteristics | Frequency | % |
|---|---|---|---|
| Gender | Male | 241 | 56.2 |
| | Female | 186 | 43.4 |
| | Non-binary | 1 | 0.2 |
| | I prefer not to say | 1 | 0.2 |
| Educational Qualification | Elementary | 10 | 2.3 |
| | Secondary | 52 | 12.1 |
| | Teacher training | 20 | 4.7 |
| | Any certification | 22 | 5.1 |
| | OND | 40 | 9.3 |
| | HND | 83 | 19.3 |
| | Bachelors | 139 | 32.4 |
| | Masters | 58 | 13.5 |
| | Doctorate | 5 | 1.2 |
| Facebook weekly use | Rarely | 58 | 13.5 |
| | Occasionally | 84 | 19.6 |
| | Sometimes | 54 | 12.6 |
| | Frequently | 107 | 24.9 |
| | Usually | 55 | 12.8 |
| | Every time | 71 | 16.6 |

Note: Facebook groups are different communities or groups in which respondents are members. OND = Ordinary National Diploma, HND = Higher National Diploma.

### 3.2. Procedure and Sampling

Respondents were recruited through a combination of purposive, convenience, and quasi-snowball sampling. An initial sample was gathered via Facebook Messenger through Nigerian contacts of the first author and public posts on Facebook intended to sample from the author's Nigerian network. In these messages, the first author asked his Facebook contacts to forward the message to their Nigerian friends on Facebook. The third author gathered an additional sample of respondents using both Facebook and LinkedIn and following a similar procedure. A final sample of respondents was gathered in person by the third author in Nigeria using a physical copy of the questionnaire.

Data were initially checked for outliers (none were found) and tested to determine whether the variables met the assumption of collinearity. The results indicated that multicollinearity fell within acceptable limits according to recommendations of Coakes and Steed (2009) and Hair (2011): (self-expression, tolerance = 0.31, VIF = 3.19; presentation of extended self, tolerance = 0.56, VIF = 1.80; social obligation, tolerance = 0.42, VIF = 2.37; social support, tolerance = 0.39, VIF = 2.55; and social interaction, tolerance, 0.66, VIF = 1.51). Although several of the independent variables (self-expression, social obligation, and social support) showed high intercorrelation (see Table 2), the collinearity statistics (tolerance and VIF) fell within acceptable standards. Residual and scatter plots showed that the assumptions of homoscedasticity, linearity, and normality were met (see Figures 1 and 2) (Pallant 2020; Hair 2011).

**Table 2.** Results of correlation analysis of independent variables.

| | Variables | N | M | SD | 1 | 2 | 3 | 4 | 5 | 6 |
|---|---|---|---|---|---|---|---|---|---|---|
| 1 | Plural Identity | 429 | 4.47 | 0.91 | - | | | | | |
| 2 | Social Obligation | 429 | 4.58 | 1.57 | 0.56 ** | - | | | | |
| 3 | Social Support | 429 | 5.03 | 1.48 | 0.58 ** | 0.65 ** | - | | | |
| 4 | Social Interaction | 429 | 5.05 | 1.41 | 0.49 ** | 0.45 ** | 0.42 ** | - | | |
| 5 | Presentation of Extended Self | 429 | 4.68 | 1.41 | 0.61 ** | 0.57 ** | 0.54 ** | 0.46 ** | - | |
| 6 | Self-Expression | 429 | 4.46 | 1.13 | 0.66 ** | 0.72 ** | 0.74 ** | 0.52 ** | 0.60 ** | - |

Note: Correlation coefficients are reported in columns "1" through "6". ** $p < 0.001$.

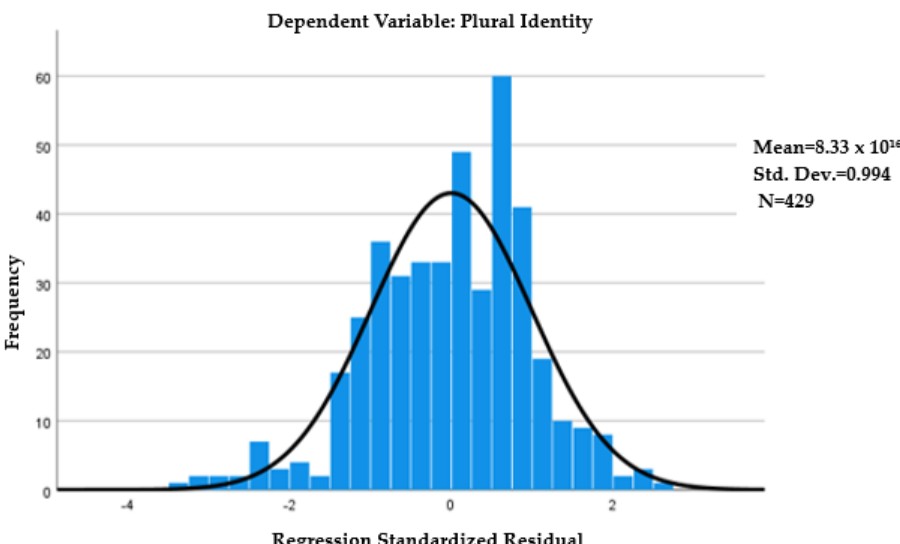

**Figure 1.** Histogram of data normality.

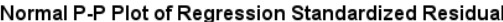
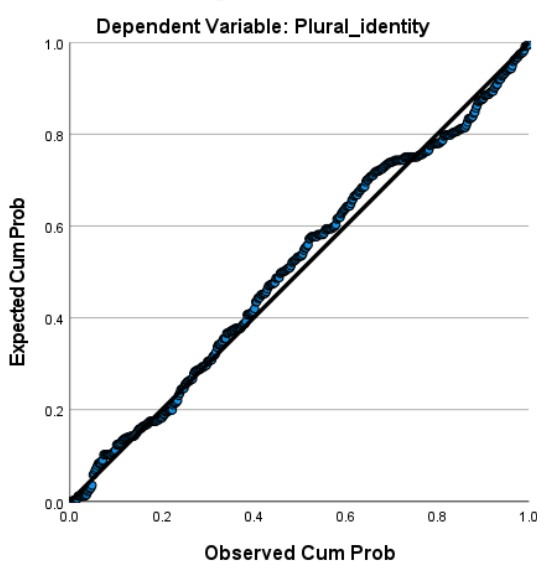

**Figure 2.** Residual plot showing data normality.

### 3.3. Measures

*Plural identity*. This dependent measure describes the multiple facets of self-identity that constitute a person's overall identity, following the conceptualization that people's identities are composed of personal and social components. Plural identity was operationalized using personal and social identity items from Nario-Redmond et al. (2004). The sixteen items were averaged to form a plural identity index. The test–retest reliability for the social identity subscale was r = 0.82, while for the personal identity subscale it was r = 0.77, $p < 0.0001$. Each item was framed to assess plural identity on social media, for example, "I am similar to people in my Facebook network" and "I identify myself as a Nigerian on Facebook". All items were measured using a 7-point Likert scale (1 = strongly disagree, 7 = strongly agree), which showed acceptable reliability Hair et al. (2019) ($\alpha = 0.80$).

*Presentation of extended self*. Belk (2013) describes the extended self as including both personal and external objects a person regards as his or her possessions, including persons, places, and groups. In other words, the extended self is an expression of possession that is personal (i.e., locatable with or within a person) and that extends beyond the physical self to external properties, including those observed on SNSs. Presentation of extended self

was measured using three items from Belk (2013) reframed for the Facebook context using a 7-point Likert scale instrument (1 = strongly disagree, 7 = strongly agree), for example, "My postings, likes, and comments on Facebook reflect that my family is important to me". This scale showed moderate reliability ($\alpha$ = 0.62) overall according to the recommendations detailed by Hair et al. (2019).

*Self-expression*. Kim and Sherman (2007) assert that self-expression is contingent on cultural context, that is, an expression of self may vary depending on culture. The core of self assumes that a person possesses a collection of internal attributes that distinguishes the individual and shapes behavior, which may be expressed via SNS. The self is expressed in different physical manifestations through behavior, and the same may be possible in virtual environments (e.g., SNSs) as well. Self-expression was measured using eight items from Galassi et al. (1974) adapted to Facebook, which were assessed using a 7-point Likert scale (1 = strongly disagree, 7 = strongly agree), and exhibited good reliability ($\alpha$ = 0.78) (Hair et al. 2019).

*Social obligation*. According to Ozanne et al. (2017), social obligation on SNSs targets messages that people think are good for others, that gratify personal morals, and that include what people want to share about themselves. Social obligation was measured using two items from Ozanne et al. (2017) adapted for Facebook. These survey questions used a 7-point Likert scale (1 = strongly disagree, 7 = strongly agree), $\alpha$ = 0.53. Although this is classified as poor reliability according to the recommendations of Hair et al. (2019), this may be due to the very limited number of items we used. However, it should be noted that a very high alpha value is not always a good indicator (Taber 2018), for instance, when multiple items are used to assess the same construct. Of additional note, Ozanne et al. (2017) did not report the reliability of their scale; thus, these measures may need to be viewed with caution.

*Social support*. Li et al. (2015) describe social support on Facebook as affordances of Facebook that support social interaction activities, which include status updating, sharing information, liking or commenting on friends' posts, and others. Two items from Sherbourne and Stewart (1991) were used to measure social support, and were adapted to the Facebook context. These were measured using a 7-point Likert scale, (1 = strongly disagree, 7 = strongly agree), $\alpha$ = 0.64, which is classified as moderate reliability (Hair et al. 2019).

*Social interaction*. According to Al-Jabri et al. (2015), social interaction refers to the ability or desire of an individual to communicate and build relationships with others. It involves social behavior where people interact and socialize with one another, and can take place on Facebook via friend requests, commenting, liking another's post, and more. We used an index for social interaction adapted from Li et al. (2015), which was measured with a 7-point Likert scale, (1 = strongly disagree, 7 = strongly agree). This index ($\alpha$ = 0.57) showed somewhat low reliability (Hair et al. 2019), though the original scale reported a Cronbach's alpha of 0.82. Accordingly, the findings from this index may require more careful consideration, similar to the social obligation index.

## 4. Results

This study investigated the relationship between personal-communicative behaviors (presentation of the extended self and self-expression) and plural identity on social media, on the one hand, and the relationship between social-communicative behaviors (social support, social obligation, and social interaction) with plural identity on the other hand. Because people's social-communicative behaviors were expected to have stronger behavioral and statistical effect on the manifestation of plural identity than personal-communicative due to gratifications related to SNSs use (Kaplan and Haenlein 2010; DiMicco and Millen 2007), a hierarchical multiple regression model was constructed that entered the social-communicative and personal-communicative variables separately.

A two-stage hierarchical multiple regression was conducted with plural identity as the dependent variable using SPSS 27. Social obligation, social support, and social interaction were entered in the first stage of the regression; age, gender, and educational qualification

were entered in the first stage alongside these three predictors to control for confounding variables. Presentation of the extended self and self-expression were entered in stage two of the regressions. The independent variables were entered in this order to account for the potentially stronger effect of social-communicative behaviors on pluralistic identity compared to personal-communicative behaviors due to differences in expected needs and gratifications using Facebook. The findings of this hierarchical multiple regression model are reported in Table 3.

**Table 3.** Results of linear regression analysis with hierarchical entry.

|  | **M1** | | | **M2** | | |
|---|---|---|---|---|---|---|
|  | **SE** | **β** | **t** | **SE** | **β** | **t** |
| Age | 0.05 | −0.18 ** | −4.85 | 0.43 | −0.16 ** | −4.50 |
| Gender | 0.06 | 0.07 | 1.80 | 0.06 | 0.06 | 1.68 |
| Educational Qualification | 0.01 | 0.12 ** | 3.21 | 0.01 | 0.09 * | 2.60 |
| Social Obligation | 0.02 | 0.27 ** | 5.55 | 0.02 | 0.09 | 1.72 |
| Social Support | 0.03 | 0.29 ** | 6.0 | 0.03 | 0.11 * | 2.13 |
| Social Interaction | 0.02 | 0.26 ** | 6.38 | 0.02 | 0.16 ** | 3.90 |
| Presentation of Extended Self |  |  |  | 0.03 | 0.25 ** | 5.64 |
| Self-Expression |  |  |  | 0.05 | 0.28 ** | 4.86 |
| $R^2$ | 0.48 |  |  | 0.55 |  |  |
| *F* for change in $R^2$ | 65.36 ** |  |  | 33.44 ** |  |  |

Note: N = 429; ** $p < 0.001$; * $p < 0.05$.

The findings from this model show that at stage one, when social-communicative behaviors of social obligation, social support, and social interaction were added alongside age, gender, and educational qualification, there was a statistically significant relationship with plural identity, $F(6, 422) = 65.36$, $p < 0.001$, which accounted for 48% of the variance in the dependent measure. Moreover, all variables of interest were statistically significant, meaning that each of the three social-communicative variables predicted plural identity in the model. The addition of the personal-communicative variables of self-expression and presentation of the extended self in stage two accounted for a 55% variance in plural identity, a change of approximately 7%. The change in $R^2$ was significant, $F(8, 420) = 64.92$, $p < 0.001$. Both of the new variables of interest introduced in stage two emerged as statistically significant. While two of the previous variables (social support and social interaction) remained significant predictors of plural identity, social obligation did not meet the traditional significance threshold in the second model ($p = 0.86$), becoming the only predicting variable of interest to not significantly predict the outcome measure.

## 5. Discussion

This article investigated five communicative behaviors that could predict the emergence of pluralistic identity in Facebook use following expectations based on the uses and gratifications paradigm by surveying a sample of active Nigerian Facebook users. Plural identity was conceptualized and measured using a combination of personal and social identity characteristics, and served as the outcome variable in the study. Predictor variables included communicative behaviors that might explain the strengthening of people's plural identity on Facebook, conceptualized and operationalized as personal-communicative behaviors (i.e., presentation of the extended self and self-expression) and social-communicative behaviors (i.e., social obligation, social support, and social interaction) along with their applicability to the SNS context. Facebook being an SNS designed to facilitate socialization, it was expected that social-communicative predictor variables would exhibit a stronger effect on plural identity than personal-communicative behaviors. Overall, our findings show that all variables relate to plural identity on Facebook except for social obligation in the scenario where personal-communicative behaviors were added

to the model. It should be noted that social obligation did predict plural identity when combined only with social-communicative behaviors; see Model 1 in Table 3. Overall, these findings indicate that identity may be navigated in digital spaces (e.g., Facebook) by engaging in both personal-communicative and social-communicative behaviors, which together combine into an overarching pluralistic identity.

Although four measured behaviors emerged as significant, the failure of social obligation to predict plural identity is explainable in two ways, i.e., statistically and behaviorally. First, the inclusion of control variables (i.e., age, gender, and educational qualification) might have influenced the relationship between social obligation and plural identity in the first model, producing a significant positive relationship. Nevertheless, the subsequent inclusion of other personal-communicative variables counteracted the initial positive relationship, resulting in a non-relationship. Second, not all SNS users may feel obligated to engage with others on social platforms. For example, people can be self-engrossed despite their presence on SNSs, while others can become passive observers without actively engaging others. Ozanne et al. (2017) emphasize that social obligations on SNSs include messages people want to share about themselves and messages that are helpful to others. While some people fulfill these two aspects of messaging, others may neglect the second component (i.e., producing and sharing messages helpful to others). In addition, it should be noted that the social obligation scale adapted from Ozanne et al. (2017) reported somewhat low reliability (Cronbach's alpha = 0.53); thus, the measures themselves could have influenced this finding.

Perhaps unexpectedly, personal-communicative behaviors exhibited a stronger relationship with plural identity on Facebook than social-communicative behaviors when comparing the beta values reported in Table 3. This might indicate that despite the social nature of SNSs, including Facebook, people are especially concerned with developing their personal identity and distinguishing themselves from other users. However, it should be noted that personal identity in social spaces is still socially constructed and enacted in relation to other people. Goffman describes personal identity as individuals' biographies, consisting of their unique human characteristics distinguishing them within social contexts (Goffman 1959). Additionally, the self and identity are produced and sustained according to societal norms, and are socially negotiated (Berger and Luckmann 2016; Burr 2003; Gergen and Gergen 2000). While scholars have separated personal identity from social identity, implying that personal and social-communicative behaviors can predict the two accordingly, the reality is that social identity connotes personal identity and vice versa. Thus, these two components of pluralistic identity seem inextricable in the behaviors of SNS users, with each contributing to an overall identity constructed through the use of the platform.

SNSs have come to stay and are now seemingly permanent storehouses for human cultural artifacts with which users interact and depend on for numerous social purposes, including identity formation. It is expected that SNSs will continue to grow over time, specifically in Africa, in accordance with the population growth on the continent (a current estimate of 1.4 billion people with average annual growth of 2% (World Population Review 2022). People are expected to increasingly turn to social platforms such as Facebook to gratify their social and communication needs, especially those needs that are unsatisfied through offline interaction.

For example, the study of Jung (2011) on identity gaps underscores the voids people may experience in offline communication that may be propellers for a concerted online presence on SNSs. In his study, Jung (2011) introduced the concept of an identity gap, and proposed three separate types: (a) personal-enacted, the difference between the self-view and the self-presented in communication; (b) personal–relational, the difference between the self-view and the perception of others' view of oneself; and (c) enacted relational, the difference between the communicated self-identity and the identity ascribed to the self by others. These communicative phenomena may create an impetus for people's continued SNS presence in seeking to satisfy identity gaps. Likewise, the article by Buckingham

(2008) on adolescent identity formation describes youths' engagement with chatting on SNSs as a space for young people to rehearse and explore aspects of identity and personal relationships that may be unavailable elsewhere. Adolescence is a stormy and stressful life stage typified by intergenerational clashes, mood swings, adventurousness, and identity crises (Hall 1972). The need to resolve these crises together with unmet offline communicative needs may propel plural identity on SNSs and its relationship with communicative behaviors.

*Limitations and Suggestions for Further Study*

There are several limitations identifiable in this study. First, the concept of plural identity explored in this study assesses identity from only two domains, namely, personal and social. Scholars have recognized additional domains of identity beyond personal and social identity; For instance, Goffman (1959) identified three domains of identity: personal, social, and ego (i.e., the psychological attributes of human identity formations). Future studies could explore this domain and other unidentified domains beyond those specified in this study. Second, the sampling procedure here was ethnically biased because the sampling was drawn only from the Nigerian population; non-probability sampling of this population is another potential limitation. Because cultural practices may differ according to groups, exploring the relationship between predictor variables and plural identity via SNS using samples from other nationalities is another area for further study.

Third, this study singularly concentrated on Facebook, though many additional social media platforms exist in the current digital ecosystem. Thus, replicating this study using other SNSs is encouraged. Finally, the somewhat low reliability of two of the indexes used in this study (social obligation and social interaction) means that certain findings should be considered cautiously. For example, the finding that social obligation did not significantly predict plural identity in Model 2 (see Table 3) could have been influenced by the measures used. Future works may want to examine social obligation more thoroughly, perhaps by developing an alternative index for this concept. The low reliability of the social interaction index, previously validated by Li et al. (2015), is more difficult to explain, though this could have resulted from cultural and/or technological changes in Facebook use subsequent to the design of this index in 2015. This possibility should be examined more closely in future work.

## 6. Conclusions

This article investigated the relationship between presentation of personal-communicative (i.e., extended self and self-expression) and social-communicative (i.e., social support, social obligation, and social interaction) behaviors with respect to plural identity using a sample of active Nigerian Facebook users. All independent variables significantly predicted plural identity on Facebook except for social obligation. Communication technologies, including SNSs, are paradoxical because scholars are divided between their benefits and downsides, generating an unresolvable polemic. On the one hand, technology provides opportunities for cohering into new communities and engaging in civic life, providing resources for individual liberation and capacities. On the other hand, it threatens privacy, introduces digital inequality, and exposes users to commercial abuse, obsession, and pornography (Buckingham 2008). Accordingly, it is imperative for technology companies to consider people's social needs and factor in robust identity features in combination with user safety as priorities in their product architectures. The socialization enabled through SNSs should be integrated with nuanced identity features in order to satisfy people's personal and social identity needs. In addition, technology companies should accentuate SNSs with more robust identity features to better satisfy people's social needs.

**Author Contributions:** Conceptualization, T.O.A.; methodology, T.O.A.; software, T.O.A.; validation, T.O.A., B.J.H., and M.I.L.; formal analysis, T.O.A. and B.J.H.; investigation, T.O.A.; resources, T.O.A.; data curation, T.O.A.; writing—original draft preparation, T.O.A.; writing—review and editing, T.O.A. and B.J.H.; visualization, T.O.A. and B.J.H.; supervision, B.J.H.; project administration, T.O.A. and

M.I.L.; funding acquisition, T.O.A. All authors have read and agreed to the published version of the manuscript.

**Funding:** This research received no external funding.

**Institutional Review Board Statement:** The study was conducted in accordance with the Declaration of Helsinki, and approved by the Institutional Review Board (or Ethics Committee) of University of Southern Mississippi (IRB-21-233, 07/19/2021). for studies involving humans.

**Informed Consent Statement:** Informed consent was obtained from all subjects involved in the study.

**Data Availability Statement:** The data presented in this study are available on request from the corresponding author. The data are not publicly available due to ethical and privacy concerns.

**Conflicts of Interest:** The authors declare no conflict of interest.

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
