# Peer review of "Understanding Motivations for Plural Identity on Facebook among Nigerian Users: A Uses and Gratification Perspective for Engaging on Social Network Sites (SNS)"

_journalmedia, doi:10.3390/journalmedia4030045_

Round 1
Reviewer 1 Report
Dear Authors,
Thank you for the opportunity to review an interesting article entitled: ‘Understanding motivations for plural identity on Facebook among Nigerian users: A uses and gratification perspective for engaging on Social Network Sites (SNS)’. The aim of this study was to analyse the relationship between the communication behaviour of Nigerian Facebook users and their multiple identities.
The strength of the article presented for evaluation is the strong literature review, as well as the author's awareness of the strengths and weaknesses of the analyses carried out.
The reviewer's job, on the other hand, is to help improve the article so that it meets the highest possible standards of the journal, therefore I will focus on its weaknesses:
General comments
[1]. Individual sections should be numbered.
[2]. Statistical test symbols should be in italics: "R2", "t", "β", etc.
Introduction
[3]. Line 268 mentions six components of 'social interaction', but then only five are referred to (line 269).
[4]. Which research questions are referred to in lines 291-292 and 303? Are these different from those mentioned in lines 248-249 and 275-276?
Method
[5]. I have doubts as to whether it is methodologically correct to combine the two modes of data collection (online and offline). What is the validity of doing so, especially as the groups are not equal.
[6]. No information is available on when the study was conducted.
[7]. There is no 'Data analysis' section where all statistical analyses used should be discussed.
[8]. When describing tools, Cronbach's α should be given for both the tool and the results from the current study.
[9]. Do the tools used in the study have Nigerian cultural adaptations? If yes, please provide this information along with the authors of the adaptations.
[10]. In line 365 'Social interaction' should be in italics.
Results
[11]. There is no indication of statistically significant correlations in Table 2.
[12]. No description of Table 2, indicating that this table lacks validity.
[13]. Lines 389-390 refer to graphs of residuals and scatter. So it would be good to insert these graphs, for a better illustration of the analyses carried out.
Bibliografia
[14]. DOI numbers should be completed in the bibliography.
Author Response
Response to Reviewers [journalmedia-2402090]
We appreciate the reviewers’ time and effort spent providing constructive feedback and appreciate the opportunity to resubmit this revised manuscript to Journalism and Media. Comments made by the reviewers are listed below, followed by our responses (in italics). Edits made to the manuscript are highlighted in text (in yellow), excepting minor typographical changes.
Reviewer 1
General comments
- Individual sections should be numbered.
Response: We have numbered the different sections as requested.
- Statistical symbols should be in italics.
Response: All statistical symbols have been italicized as requested.
Introduction
- Line 268 mentions six components of 'social interaction', but then only five are referred to (line 269).
Response: Thank you for noticing this. The sixth component, context, has been added.
- Which research questions are referred to in lines 291-292 and 303? Are these different from those mentioned in lines 248-249 and 275-276?
Response: We apologize for the confusion! The secondary clauses referring to the RQs have been removed.
Method
- I have doubts as to whether it is methodologically correct to combine the two modes of data collection (online and offline). What is the validity of doing so, especially as the groups are not equal.
Response: We maintain that the offline data collected is valid to the research objectives because we inserted screening questions on the questionnaire both online and offline to weed away invalid responses. Examples include “Are you a Nigerian citizen?” “How frequent do you use Facebook in a week?” and “To how many Facebook groups do you belong?” Moreover, the two samples exhibited no noteworthy differences, and thus the additional statistical power afforded by merging the datasets seemed worthwhile.
- No information is available on when the study was conducted.
Response: The data was collected between July 2021 and January 2022. This has been added under section 3.1 about participants and data.
- There is no 'Data analysis' section where all statistical analyses used should be discussed.
Response: Information about our analysis procedures are provided in the results section at length, and thus we felt that adding another section to methods would create redundancy. However, if the reviewer would like us to add this section, we would be happy to comply.
- When describing tools, Cronbach's α should be given for both the tool and the results from the current study.
Response: Alphas have been reported in the methods section for each index. It should be noted here that two scales were less reliable than in previous applications (though one scale did not have previously reported alphas), which has been described in some length across the method, discussion, and limitations sections.
- Do the tools used in the study have Nigerian cultural adaptations? If yes, please provide this information along with the authors of the adaptations.
Response: Since the use of social media, Facebook in this case, is ubiquitous, there were no particular Nigerian cultural adaptations, except in the framing of the items, where necessary, to make them relevant to the subjects investigated. We have added relevant examples of these items to the manuscript.
- In line 365 'Social interaction' should be in italics.
Response: Thank you for catching this. It has been corrected.
Results
- There is no indication of statistically significant correlations in Table 2
Response: P-values have been added to Table 2 alongside the correlation coefficients.
- No description of Table 2, indicating that this table lacks validity.
Response: A description has been added to Table 2 in addition to p-values. Additionally, an in-text passage describes Table 2’s correlations alongside tests of multicollinearity (p. 20).
- Lines 389-390 refer to graphs of residuals and scatter. So it would be good to insert these graphs, for a better illustration of the analyses carried out.
Response: A histogram and residual plot have been added to the manuscript to show linearity and normality. However, the scatter plots were not added due to the amount of space they would require (there are six scatter plots to display). If the reviewer feels strongly about including these plots, and the editor is okay with the space they would require, we would be happy to include these.
Bibliografia
- DOI numbers should be completed in the bibliography.
Response: We included available DOI numbers for the references in this version.

Reviewer 2 Report
The article is about plural identity formations of Nigerian Facebook users. First, the concept of plural identity is introduced along with its related definitions. The relationship between the communicative behaviors of users on social network sites (SNS) and their plural identities is revealed as the main point of discussion. The uses and gratifications perspective on SNS is evaluated for a specific sample using survey-analysis. Even though there is already plenty of academic work available in the field, communicative behaviors of users in relation to the limitations and affordances of SNS is still a contemporary topic of interest.
It would be more objective to have a more random sample for this research. Rather than sending the initial invitation to friends in researchers` networks, a public announcement could have been made. The F2F survey procedure may help or even worsen the randomization in this case.
I would also suggest a more thorough analysis of extended selves; especially given the fact that Facebook users are now much older and their use patterns vary. H. Kang (2021) has a recent work in this regard which may be helpful.
Good to go with minor revisions.
Author Response
Response to Reviewers [journalmedia-2402090]
We appreciate the reviewers’ time and effort spent providing constructive feedback and appreciate the opportunity to resubmit this revised manuscript to Journalism and Media. Comments made by the reviewers are listed below, followed by our responses (in italics). Edits made to the manuscript are highlighted in text (in yellow), excepting minor typographical changes.
Reviewer 2
Comments
The article is about plural identity formations of Nigerian Facebook users. First, the concept of plural identity is introduced along with its related definitions. The relationship between the communicative behaviors of users on social network sites (SNS) and their plural identities is revealed as the main point of discussion. The uses and gratifications perspective on SNS is evaluated for a specific sample using survey-analysis. Even though there is already plenty of
academic work available in the field, communicative behaviors of users in relation to the limitations and affordances of SNS is still a contemporary topic of interest.
It would be more objective to have a more random sample for this research. Rather than sending the initial invitation to friends in researchers` networks, a public announcement could have been
made. The F2F survey procedure may help or even worsen the randomization in this case.
Response: We appreciate this observation. We agree that a random sampling method would have been preferable for affording generalizability. However, a convenience sample was used in this case to capture the target demographic. We did supplement the online survey with F2F collection to allow some degree of triangulation in case differences emerged. As mentioned to reviewer 1, no noteworthy differences emerged between these samples. Regardless, in future studies we would like to design and employ a more random sampling procedure.
I would also suggest a more thorough analysis of extended selves; especially given the fact that Facebook users are now much older and their use patterns vary. H. Kang (2021) has a recent work in this regard which may be helpful.
Response: We thank the reviewer for this suggestion. In the literature review section, under “presentation of extended self,” we added an overview of Kang and Shin (2021) that we think helps bolster our review: “Kang and Shin (2021) investigated the mediated relationship between the use of Facebook as extended self and users’ privacy management online. They found that motivations of self-expression, belonging, and archiving personal memory were indirectly associated with privacy management. Their findings underscore the proposition that self-extension to Facebook is a vital psychological construct determining how various self-related incentives impact users’ privacy management.”

Round 2
Reviewer 1 Report
I appreciate the authors' efforts to improve the manuscript. However:
1) In the 'Results' section, only the results obtained and their statistical analyses should be presented. The place to describe the procedures used is in the "Data analysis" section, which should be in the "Method" section.
2) When describing the scales used, redundant information on mean (M) and standard deviation (SD) and test-retest scores is given.
3) The fact of Facebook's ubiquity is not a sufficient rationale for using scales that do not have proven psychometric properties for a given country and culture.
4) Table 2 still does not indicate which correlations are statistically significant.
5) There are still many bibliographic items without DOI numbers and in some cases no bibliographic data (e.g. item two: Al-Jabri, I. M., Sohail, M. S., & Ndubisi, N. O. (2015). Understanding the usage of global social networking sites by Arabs through the lens of uses and gratifications theory. Journal of Service Management.).
Author Response
Response to reviewer 3
Thank you for giving us the opportunity to revise and resubmit our manuscript for publication in Journalism and Media. Please, do not hesitate to let us know if you want us to make further revisions. Please, find our italicized responses after each of your comments/concerns below:
1) In the 'Results' section, only the results obtained and their
statistical analyses should be presented. The place to
describe the procedures used is in the "Data analysis"
section, which should be in the "Method" section.
Response:
We moved this paragraph with relevant diagrams to method section:
Data were initially checked for outliers (none were found) and tested to determine if variables met the assumption of collinearity. The result indicated that multicollinearity fell within acceptable limits, according to recommendations from Coakes and Steed (2009) and Hair (2011): (self-expression, tolerance = .31, VIF = 3.19; presentation of extended self, tolerance = 0.56, VIF = 1.80; social obligation, tolerance = 0.42, VIF = 2.37; social support, tolerance = 0.39, VIF = 2.55; and social interaction, tolerance, 0.66, VIF = 1.51). Although some of the independent variables (self-expression, social obligation, and social support) showed high intercorrelation (see Table 2), the collinearity statistics (tolerance and VIF) fell within acceptable standards. Residual and scatter plots showed that the assumptions of homoscedasticity, linearity, and normality were also met (see Figures 1 and 2) (Pallant, 2020; Hair, 2011).
Table 2: Results of correlation analysis of independent variables
|
|
Variables |
N |
M |
SD |
1 |
2 |
3 |
4 |
5 |
6 |
|
1 |
Plural Identity |
429 |
4.47 |
.91 |
- |
|
|
|
|
|
|
2 |
Social Obligation |
429 |
4.58 |
1.57 |
.56** |
- |
|
|
|
|
|
3 |
Social Support |
429 |
5.03 |
1.48 |
.58** |
.65** |
- |
|
|
|
|
4 |
Social Interaction |
429 |
5.05 |
1.41 |
.49** |
.45** |
.42** |
- |
|
|
|
5 |
Presentation of Extended Self |
429 |
4.68 |
1,41 |
.61** |
.57** |
.54** |
.46** |
- |
|
|
6 |
Self-Expression |
429 |
4.46 |
1.13 |
.66** |
.72** |
.74** |
.52** |
.60** |
- |
Note: Correlation coefficients are reported in columns “1” through “6.” ** p <.001
Figure 1: Histogram of data normality
Figure 2: Residual plot showing data normality.
2) When describing the scales used, redundant information on
mean (M) and standard deviation (SD) and test-retest
scores is given.
Response:
We removed the mean (M) and standard deviation (SD) information from measure subsection, except for plural identity sub-subsection of measures where Nario-Redmond et al. (2004) reported test-retest result as the psychometric properties for their scale.
3) The fact of Facebook's ubiquity is not a sufficient rationale
for using scales that do not have proven psychometric
properties for a given country and culture.
Response:
We provided the psychometric properties of the original items and the ones we eventually used in the measurement subsection, except for social obligation adapted from Ozanne et al. (2017) where the authors did not report their scale reliability.
4) Table 2 still does not indicate which correlations are
statistically significant.
Response:
Apologies about this. We did it in our last revision but mistakenly submitted the wrong version. All the correlations are statistically significant as indicated in this revision.
5) There are still many bibliographic items without DOI
numbers and in some cases no bibliographic data (e.g. item
two: Al-Jabri, I. M., Sohail, M. S., & Ndubisi, N. O. (2015).
Understanding the usage of global social networking sites
by Arabs through the lens of uses and gratifications theory. Journal of Service Management.).
Response:
We provided bibliographic data for item
two: Al-Jabri, I. M., Sohail, M. S., & Ndubisi, N. O. (2015). Also, we provided Doi information for other items in the reference section, except for those we were unable to find.
